# Urinary Epidermal Growth Factor/Creatinine Ratio and Graft Failure in Renal Transplant Recipients: A Prospective Cohort Study

**DOI:** 10.3390/jcm8101673

**Published:** 2019-10-13

**Authors:** Manuela Yepes-Calderón, Camilo G. Sotomayor, Matthias Kretzler, Rijk O. B. Gans, Stefan P. Berger, Gerjan J. Navis, Wenjun Ju, Stephan J. L. Bakker

**Affiliations:** 1Division of Nephrology, Department of Internal Medicine, University Medical Center Groningen, University of Groningen, 9700 RB Groningen, The Netherlands; manueyepes@gmail.com (M.Y.-C.); s.p.berger@umcg.nl (S.P.B.); g.j.navis@umcg.nl (G.J.N.); s.j.l.bakker@umcg.nl (S.J.L.B.); 2Department of Internal Medicine, Department of Computational Medicine and Bioinformatics, University of Michigan, Ann Arbor, MI 48109, USA; kretzler@umich.edu (M.K.); wenjunj@med.umich.edu (W.J.); 3Department of Internal Medicine, University Medical Center Groningen, University of Groningen, 9700 RB Groningen, The Netherlands; r.o.b.gans@umcg.nl

**Keywords:** epidermal growth factor, creatinine, graft failure, renal transplantation.

## Abstract

Graft failure (GF) remains a significant limitation to improve long-term outcomes in renal transplant recipients (RTR). Urinary epidermal growth factor (uEGF) is involved in kidney tissue integrity, with a reduction of its urinary excretion being associated with fibrotic processes and a wide range of renal pathologies. We aimed to investigate whether, in RTR, uEGF is prospectively associated with GF. In this prospective cohort study, RTR with a functioning allograft ≥1-year were recruited and followed-up for three years. uEGF was measured in 24-hours urine samples and normalized by urinary creatinine (Cr). Its association with risk of GF was assessed by Cox-regression analyses and its predictive ability by C-statistic. In 706 patients, uEGF/Cr at enrollment was 6.43 [IQR 4.07–10.77] ng/mg. During follow-up, 41(6%) RTR developed GF. uEGF/Cr was inversely associated with the risk of GF (HR 0.68 [95% CI 0.59–0.78]; *P* < 0.001), which remained significant after adjustment for immunosuppressive therapy, estimated Glomerular Filtration Rate, and proteinuria. C-statistic of uEGF/Cr for GF was 0.81 (*P* < 0.001). We concluded that uEGF/Cr is independently and inversely associated with the risk of GF and depicts strong prediction ability for this outcome. Further studies seem warranted to elucidate whether uEGF might be a promising marker for use in clinical practice.

## 1. Introduction

Although in recent decades short-term graft survival has seen great improvement, chronic graft failure remains a major clinical challenge for renal transplantation with no significant reduction achieved in the same time frame [1]. Graft failure is a culmination of several factors, including chronic rejection, toxicity of calcineurin inhibitors, infection, hypertension, oxidative stress, and proteinuria, together leading to progressive fibrosis and loss of renal function [2,3,4,5]. In clinical settings, most biomarkers used for follow-up, e.g., urinary albumin excretion and urinary protein excretion, are indicators of glomerular damage [6], improper of the development of fibrosis, which is an early event in the natural history of chronic rejection [3]. Finding non-invasive biomarkers that could reflect the pathophysiological changes in the renal tissue would be of remarkable utility as potential tools to monitor patients and timely identify those at high risk of graft failure [7], who could benefit from further interventions and stricter follow-up before structural damage is already present [8].

Epidermal growth factor (EGF) is a 53-amino acid peptide produced in the kidney at the ascending loop of Henle and the distal convoluted tubule [7,8]. It stimulates the proliferation and differentiation of epidermal and epithelial cells, and under normal conditions it has a critical role in renal development [7], maintenance of renal tubule integrity and tubular regenerative response to acute kidney injury [9,10,11]. However, the dysregulation and chronic activation of its receptor is known to promote pro-inflammatory response [12]; furthermore, it has been implicated in the development of interstitial fibrosis [13]. In clinical settings, the urinary excretion of EGF has shown to be decreased in a wide range of kidney pathologies—e.g., diabetic nephropathy and IgA nephropathy—suggesting that it could potentially work as a biomarker of a pathway which is common to several kidney tissue insults [14]. Although it would not be possible to summarize the complexity of the graft failure process with one biomarker, fibrosis is an important step towards graft failure development [2]; and suppression of urinary EGF (uEGF) is an early marker of this phenomenon [15]. It may be theorized that uEGF could also be altered in patients at high risk of graft failure; however, the potential association with outcome or predictive ability of uEGF for graft failure is yet to be evaluated. 

In the current study, we aimed to investigate the hypothesis that uEGF is prospectively associated with the risk of graft failure in a large, well-phenotyped, cohort of stable renal transplant recipients (RTR). Furthermore, we aimed to evaluate the prediction ability of uEGF for graft failure.

## 2. Materials and Methods

### 2.1. Study Design and Patient Population

In this prospective cohort study, all adult RTR with a functioning graft for ≥1 year, without history of drug addiction, alcohol addiction or malignancy, who visited the outpatient clinic of the University Medical Center of Groningen (The Netherlands) between November 2008–May 2011 were invited to participate. In total 707 (86%) of the 817 eligible RTR signed a written informed consent. RTR with missing information about uEGF at enrollment (*n* = 57) were excluded, resulting in 649 RTR eligible for the statistical analyses (Figure 1). There were no significant differences in risk factors for graft failure between patients with complete data and patients with missing data (Appendix A). The primary end point of the current study was death-censored graft failure, defined as restart of dialysis or need of re-transplantation. The patients were followed-up for a total of 3 years. We contacted general practitioners or referring nephrologists in cases where the status of a patient was unknown. No participants were lost to follow-up (Figure 1). The current study was approved by the institutional review board (METc 2008/186) and adhered to the Declarations of Helsinki and Istanbul.

### 2.2. Data Collection

Data at enrollment were collected during a visit to the outpatient clinic, following a detailed protocol described elsewhere [16,17]. Systolic blood pressure (SBP) and diastolic blood pressure (DBP) were measured using a semiautomatic device (Dinamap 1846, Critikon, Tampa, Florida, USA) every minute for 15 minutes, following a strict protocol as described before [16].

Other relevant donor, recipient, and transplant information was extracted from the Groningen Renal Transplant Database [18]. Delayed graft function was defined as oliguria for 7 days or need for continuous ambulatory peritoneal dialisys or need for >2 sessions of hemodyalisis. Data collection is ensured by the continuous surveillance system of the outpatient clinic of our university hospital and close collaboration with affiliated hospitals.

### 2.3. Laboratory Measurements and Calculations

According to a strict protocol, all RTR were asked to collect a 24-hours urine sample during the day before to their visit to the outpatient clinic and on that day fasting blood samples were taken. Serum creatinine was determined using the Jaffé reaction (MEGA AU510, Merck Diagnostica, Germany); plasma glucose by the glucose oxidase method (YSI 2300 Stat Plus, Yellow Springs Instruments, Yellow Springs, OH, USA). uEGF concentration was measured by ELISA (R&D Systems, Minneapolis, MN, USA); the test has a range of detection of 3.9–250 pg/mL and the intra- and inter-plate coefficients of variation were less than 10% and 15%, respectively [15]. Urinary creatinine concentration was measured by colorimetric detection kit (Enzo, New York, NY, USA). Finally, the concentration of uEGF was normalized by the concentration of urinary creatinine, and a ratio was created and used for all analyses (uEGF/Cr).

Body surface area was calculated according to the Du Bois formula [19], estimated glomerular filtration rate (eGFR) by the serum creatinine based Chronic Kidney Disease EPIdemiology collaboration equation (CKD-EPI) [20] and the cumulative dose of prednisolone as the sum of the maintenance dose of prednisolone from transplantation until enrollment.

### 2.4. Statistical Analysis 

Data analyses, computations, and graphs were performed with SPSS 22.0 software (IBM Corporation, Chicago, IL, USA) and GraphPad Prism version 7 software (GraphPad Software, San Diego, CA, USA). Descriptive statistics data are presented as mean ± standard deviation (SD) for normally distributed data, and as median (interquartile range [IQR]) for variables with a non-normal distribution. Categorical data are expressed as number (percentage). Differences in characteristics at enrollment between patients with and without data on uEGF, and among subgroups of RTR according to tertiles of uEGF/Cr were tested by one-way ANOVA for continuous variables with normal distribution, Mann–Whitney U test for continuous variables with skewed distribution and χ^2^ test for categorical variables. We also performed linear regression analyses testing the association between time after transplantation and uEGF/Cr in crude and multivariable analyses with adjustment for use of cyclosporine inhibitors. For all statistical analyses, a statistical significance level of *P* ≤ 0.05 (two-tailed) was used.

Graft failure development was visualized by Kaplan-Meier curves according to tertiles of uEGF/Cr, with statistical significance among curves tested by log-rank (Mantel–Cox) test. The prospective association of uEGF/Cr with risk of graft failure during follow-up was further examined, incorporating time to event, by means of uni- and multivariate Cox proportional-hazards regression analyses with time-dependent covariates to calculate hazard ratios (HR) and 95% confidence intervals (CI). First, we performed an unadjusted model. Afterwards we adjusted for age and sex, and the following variables: in model 2, transplant related data (transplant vintage, pre–emptive transplantation, age and sex of donor, type of donor and cold ischemia time); in model 3, renal transplant recipient characteristics (human leukocyte antigen [HLA] mismatch with donor and delayed graft function); in model 4, we adjusted for the variables included in the model 2 and 3; in model 4, immunosuppressive therapy (usage of calcineurin inhibitors and proliferation inhibitors, and acute rejection treatment); in model 5, graft function (eGFR and urinary protein excretion); and the final model (model 6) was a combination of model 4 and 5. Schoenfeld residuals were calculated to assess whether proportionality assumptions were fulfilled. Furthermore, we tested the potential predictive ability of uEGF/Cr for graft failure by means of performing a receiver operating characteristics (ROC) curve. To investigate whether uEGF might be of additional value to urinary albumin excretion and protein excretion, we calculated the individual C-statistic of these variables, and then the C-statistic of them combined with uEGF/Cr. Moreover, we performed an F-test to check whether the difference between predictive models was significant. Positive and negative predictive value were calculated for the cut-off points of the uEGF/Cr tertiles. 

As secondary analyses, we assessed potential effect-modifications by pre-specified variables of: age, sex, eGFR, plasma creatinine concentration, proteinuria, high-sensitivity C-reactive protein (hs-CRP), acute rejection, and transplantation without dialysis (pre-emptive) by fitting models containing both main effects and their cross-product terms. Finally, we performed sensitivity analyses in which we eliminated patients with extreme values of uEGF/Cr (outside −2 and 2 standard deviations).

## 3. Results

### 3.1. Characteristics at Enrollment

In total 649 RTR were included in the analyses with a mean ± SD age of 53 ± 13 years, 57% men. Patients were included at a median (IQR) of 5.28 (1.74–12.00) years after transplantation and uEGF/Cr ratio had a median of 6.43 (4.07–10.77) ng/mg. In crude linear regression analyses, there was no significant association between years after transplantation and uEGF/Cr (Std. β = −0.015; *P* = 0.71), however, the association became apparent after the adjustment for calcineurin inhibitors usage (Std. β = −0.81; *P* = 0.046). Characteristics at enrollment of the overall RTR population and according to tertiles of uEGF/Cr are shown in Table 1. In the highest uEGF/Cr tertile patients had older age (*P* = 0.01), smaller percentage of male population (*P* < 0.001), higher eGFR (*P* < 0.001), lower urinary protein excretion (*P* < 0.001), larger percentage of transplant from living donors (*P* < 0.001), younger donors age (*P* < 0.001), and higher percentage of donors were male (*P* = 0.03). Also, they used less cyclosporine (*P* = 0.002) and tacrolimus (*P* < 0.001) in their immunosuppressive regimens, but more mycophenolic acid (*P* = 0.03); and a smaller percentage of patients required acute rejection treatment (*P* < 0.001) (Table 1). Patients in the highest uEGF/Cr tertile also had higher glycated hemoglobin percentage (Table 1), independent of whether they were diabetic or non-diabetic subjects (Appendix A). 

### 3.2. Prospective Analyses on Graft Failure

During a follow-up of 3 years, 41 (6%) RTR developed graft failure. Thirty-three events (80%) were in the lowest tertile of uEGF/Cr, 4 (10%) in the intermediate tertile and 4 (10%) in the highest tertile. The curves were significantly different according to the log-rank (Mantel cox) test (*P* < 0.001). The corresponding Kaplan–Meier curves are shown in Figure 2. 

Cox regression analyses showed that uEGF/Cr ratio is inversely associated with the risk of graft failure (HR 0.68 [95% CI 0.59‒0.78] per ng/mg) and this association is highly significant (*P* < 0.001). Further adjustment for transplantation-related data, renal transplant recipient characteristics, immunosuppressive therapy, eGFR and urinary protein excretion did not materially change this finding. The association between uEGF/Cr and graft failure was still strongly significant in the final model which included adjustment for both immunosuppressive therapy and graft function, with a HR of 0.79 (95% CI 0.67‒0.94; *P* = 0.007) (Table 2).

A ROC curve assessing the prediction ability of uEGF/Cr for graft failure is displayed in Figure 3. uEGF/Cr showed to be a good predictor of the development of graft failure up to the following three years (C-statistic = 0.81), with better predictive ability than urinary albumin excretion and urinary protein excretion (C-statistic = 0.78 and C-statistic = 0.76, respectively). The curve of uEGF/Cr was significantly different from the reference line (*P* < 0.001). Being on the first tertile of uEGF/Cr had a positive predictive value of 75% for the development graft failure, on the other hand, being in the third tertile had a negative predictive value of 81% (Appendix A).

Urinary protein excretion and urinary albumin excretion had a C-statistic of 0.76 and 0.78, respectively. The predictive value for both variables was significantly improved after the addition of uEGF/Cr (C-statistic = 0.82, F-test for difference among models = *P* < 0.001) (Table 3).

### 3.3. Secondary and Sensitivity Analysis 

In effect-modification analyses we found that none of the pre-specified variables we explored (age, sex, eGFR, plasma creatinine concentration, proteinuria, hs-CRP, acute rejection, and pre-emptive transplantation) was a significant effect-modifier of the association between uEGF/Cr and the risk of graft failure (*P* > 0.10), therefore we did not proceed with any subgroup analyses (Appendix A).

Finally, in the sensitivity analyses in which we removed patients with extreme values of uEGF/Cr (patients outside of the −2 and +2 standard deviation), our findings remained materially unchanged. uEGF/Cr was strongly inversely associated with risk of graft failure (HR 0.68 (95% CI 0.59–0.78); *P* < 0.001) and further adjustments analogous to models used in the primary analyses did not materially modified this association (Appendix A).

## 4. Discussion

In a large cohort of stable RTR, we showed first, that patients with impaired renal function have significantly lower excretion of uEGF. Second, that uEGF/Cr is inversely associated with the risk of graft failure and that this association is independent of potential confounders, including immunosuppressive therapy, eGFR and urinary protein excretion. Finally, uEGF/Cr also appears to have good prediction ability for the development of graft failure, superior to urinary albumin excretion and urinary protein excretion. These findings are in agreement with previous evidence showing that uEGF is a biomarker altered in several kidney pathologies [15,21,22], and for the first time we provided evidence in the post-renal transplantation setting. 

EGF is a 53-amino acid peptide which expression is restricted to the kidney [7,15,22], particularly to the thick ascending limb of Henle and the distal tubule [14], therefore it is found in higher concentrations in urine than in any other body fluid [23]. EGF and its receptor are involved in several processes within kidney tissue, mainly related to tubular cell proliferation [13] and pathways of cell survival [10,11], making EGF a critical component in promoting kidney recovery from acute injury [11]. Therefore, its dysregulation is involved in key pathogenic pathways that drive kidney disease progression independent of etiology, e.g., chronic inflammation [24], extracellular matrix modulation and tubular cell dedifferentiation [15].

EGF has gained interest as a biomarker of renal disease because its decreased urinary excretion has been observed in nearly all rodent kidney injury models [20] and in various human kidney diseases [25], including diabetic nephropathy, IgA nephropathy, and lupus nephritis [14]. Consistently, we found that our study population of RTR had a decreased uEGF/Cr ratio when compared to healthy subjects, and comparable ratios to those of patients with chronic kidney disease [15,26,27]. Its common clinical standardization by creatinine (uEGF/Cr) has shown several advantages as a biomarker of kidney tissue damage: (i) it is highly tissue specific, which makes it robust to extra renal events that may affect the accuracy of other nonspecific biomarkers; (ii) it is known that even in the normal creatinine range there is a significant influence of kidney function on uEGF/Cr [22]; and (iii) it shows only a weak correlation with markers of glomerular damage as urinary protein excretion, which shows that uEFG/Cr is a representation of a different independent pathophysiologic mechanism [12,14,27] and could complement these other parameters. Our study further supports the role of uEGF/Cr as a biomarker of damage to renal tissue, and more importantly, as a biomarker independently associated with risk of graft failure in stable renal transplant recipients. Furthermore, the strong prediction abilities of uEGF/Cr for risk of graft failure, even superior to those of urinary albumin excretion and urinary protein excretion, and of adding predictive value in combination with these variables, also supports the idea of uEGF/Cr being a marker of a different pathological aspect of graft failure which might be earlier than stablished glomerular damage.

Because risk of graft failure increases with time, one could speculate that uEGF decreases with time after transplantation. However, we did not observe such a relationship over increasing tertiles of uEGF. This finding may be explained by a confounding effect of use of cyclosporine resulting in lowering of uEGF, which is supported by the observation that an association between uEGF and time after transplantation became apparent after adjustment for use of cyclosporine in linear regression analyses. We also found in our population that the use of calcineurin inhibitors was higher among patients with lower uEGF/Cr. This is in agreement with previous studies showing an inverse association between uEGF and the use of calcineurin inhibitors [28,29] and a potential involvement of the EGF receptor in the alterations that lead to magnesium loss in renal transplant recipients receiving calcineurin inhibitors [29,30]. Nevertheless, the association between uEGF/Cr and graft failure was independent of the adjustment for the use of calcineurin inhibitors. This suggests that the association of uEGF/Cr is not mediated by a nephrotoxic effect of calcineurin inhibitors, but is mediated by other mechanisms, which may involve renal fibrosis. Furthermore, in contrast to previous results [31], no difference was observed in the prevalence of diabetes between the uEGF/Cr tertiles in our study population. 

The present study has several strengths. We assessed not only the association but also the risk-prediction ability for graft failure of uEGF/Cr. Also, our extensively phenotyped cohort allowed us to control for several potential confounders, among which demographic and anthropometric variables, renal function and immunosuppressive therapy were accounted for. The following limitations should be considered in the interpretation of our results. This study was carried out in a single center with over-representation of Caucasian population, which calls prudence to extrapolation of our results to different populations regarding ethnicity. Also, we did not have repeated uEGF measurements, and the single measurement of the variable of interest could have given rise to the underestimation of the true effect [32,33]. Moreover, we used the Jaffé method to measure serum creatinine, which can generate false positive results in the presence of pesudochromogens such as ketones [34]. Next, only limited data were available regarding donors characteristics and therefore we could not adjust for donor variables such as donor serum creatinine or donor hepatitis C status. Finally, as with any observational study, residual confounding may occur despite the substantial number of potentially confounding factors for which we adjusted.

## 5. Conclusions

uEGF/Cr is inversely and independently associated with the risk of graft failure in stable RTR. This study provides for the first time relevant prospective data on a potential role of EGF in the pathophysiological changes that lead to graft failure. Furthermore, it appears that uEGF/Cr could be a biomarker of interest in the identification of patients at high risk of graft failure. Of note, to the best of our knowledge, current reference values for uEGF/Cr have not been established. Given our findings standardized assays for uEGF with reference values being generated are warranted. The potential utility of EGF directed therapies or the implementation of uEGF/Cr in clinical care of stable RTR requires further research and validation in a larger and more heterogeneous clinical studies.

## Figures and Tables

**Figure 1 jcm-08-01673-f001:**
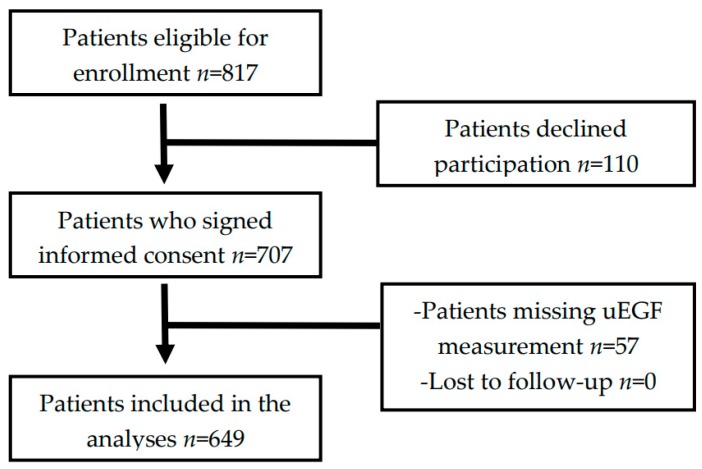
Participant flow diagram.

**Figure 2 jcm-08-01673-f002:**
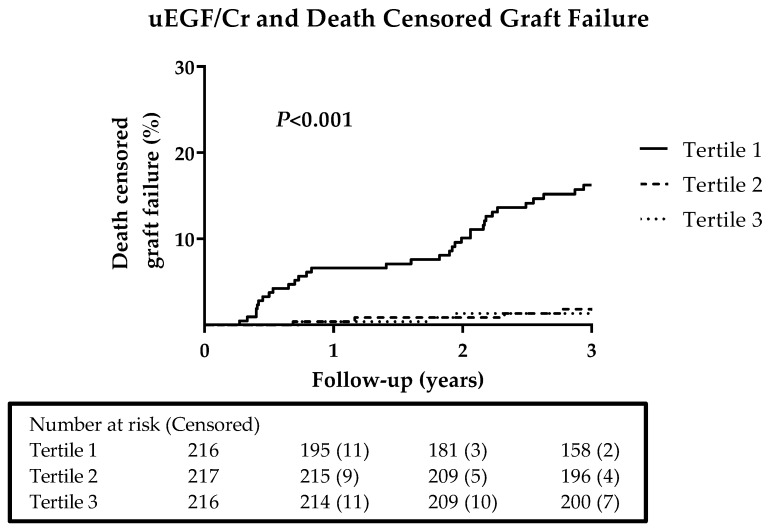
Kaplan–Meier curves by tertiles of uEGF/Cr on graft failure. Tertile 1: < 4.78 ng/mg; Tertile 2: 4.78–8.80 ng/mg; Tertile 3: > 8.80 ng/mg. *P* value was obtained from the log-rank (Mantel cox) test. uEGF/Cr, urinary epidermal growth factor/creatinine ratio.

**Figure 3 jcm-08-01673-f003:**
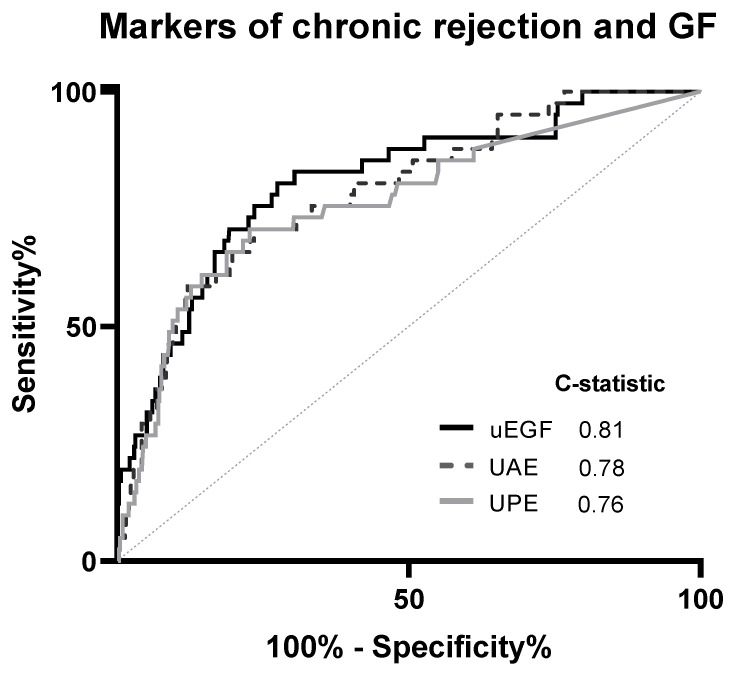
ROC curve of uEGF/Cr for graft failure. During a follow-up of 3 years, 41 (6%) patients developed graft failure. GF, graft failure; uEGF/Cr, urinary epidermal growth factor/creatinine ratio; UAE, urinary albumin excretion; UPE, urinary protein excretion.

**Table 1 jcm-08-01673-t001:** Characteristics at enrollment of the study population.

Characteristics	Overall RTR (*n* = 649)	Tertile 1	Tertile 2	Tertile 3	*P*
		<4.78 ng/mg	4.78–8.80 ng/mg	>8.80 ng/mg	
uEGF/Cr, ng/mg	6.43 (4.07–10.77)	3.18 (2.12–4.07)	6.43 (5.57–7.45)	12.91 (10.77–16.08)	—
Demographics					
Age, years	53 ± 13	51 ± 13	53 ± 13	55 ± 12	0.01
Sex (male), *n* (%)	373 (57)	149 (69)	124 (57)	100 (46)	<0.001
Caucasian ethnicity, *n* (%)	647 (99)	216 (100)	216 (100)	215 (99)	0.44
Renal allograft function					
eGFR, mL/min/1.73 m^2 a^	52 ± 20	37 ± 14	53 ± 16	68 ± 17	<0.001
Urinary protein excretion, g/24 h ^b^	0.20 (0.02–0.34)	0.25 (0.13–0.63)	0.19 (0.02–0.32)	0.08 (0.02–0.26)	<0.001
Urinary albumin excretion, mg/24 h ^c^	38.27 (10.57–174.38)	94.00 (20.48–393.77)	37.52 (10.50–155.20)	19.35 (7.11–71.08)	<0.001
Renal transplantation characteristics					
Pre–emptive transplantation, *n* (%)	105 (16)	27 (13)	35 (16)	43 (20)	0.11
Living donor, *n* (%) ^d^	230 (35)	30 (14)	95 (44)	75 (35)	0.002
Age of donor, years ^e^	43 ± 15	47 ± 14	45 ± 15	37 ± 15	<0.001
Sex of donor (male), *n* (%) ^f^	331 (51)	97 (45)	109 (50)	125 (58)	0.03
Cold ischemia time, hours ^g^	15.2 (2.7–21.3)	15.6 (3.0–22.5)	14.0 (2.6–21.0)	15.4 (2.7–22.0)	0.03
Time since transplantation, years	5.28 (1.74–12.00)	5.07 (1.53–12.92)	5.26 (1.40–12.32)	5.45 (2.63–10.98)	0.96
Renal transplantation recipients’ characteristics					
Delayed graft fuction, *n* (%)	47 (7)	24 (11)	13 (6)	10 (5)	0.02
HLA mismatch with donor, number ^h^	2 (1–3)	2 (1–3)	2 (1–3)	2 (1–3)	0.10
Immunosuppressive therapy					
Cumulative prednisolone dose, g	17.4 (5.2–36.2)	17.0 (4.7–38.4)	16.8 (4.6–36.4)	18.1 (8.1–32.8)	0.78
Sirolimus or rapamune use, *n* (%) ^i^	13 (2)	4 (2)	6 (3)	3 (1)	0.57
Type of calcineurin inhibitor
Cyclosporine, *n* (%)	258 (40)	90 (42)	102 (47)	66 (31)	0.002
Tacrolimus, *n* (%)	120 (18)	66 (31)	32 (15)	22 (10)	<0.001
Type of proliferation inhibitor					
Mycophenolic acid, *n* (%)	424 (65)	126 (58)	147 (68)	151 (70)	0.03
Azathioprine, *n* (%)	112 (17)	41 (19)	32 (15)	39 (18)	0.47
Acute rejection treatment, *n* (%)	172 (27)	77 (36)	55 (25)	40 (19)	<0.001
Body composition					
Body surface area, m^2^	1.94 ± 0.22	1.97 ± 0.23	1.94 ± 0.20	1.92 ± 0.21	0.06
Body mass index, kg/m^2^	26.5 ± 4.7	26.4 ± 4.8	26.4 ± 4.4	26.8 ± 4.9	0.53
Cardiovascular history					
History of cardiovascular disease, *n* (%) ^j^	281 (43)	88 (41)	96 (44)	97 (45)	0.65
Arterial pressure					
SBP, mmHg ^a^	136 ± 17	138 ± 18	135 ± 17	135 ± 17	0.17
DBP, mmHg ^a^	82 ± 11	84 ± 11	82 ± 11	82 ± 10	0.06
Use of antihypertensives, *n* (%)	573 (88)	202 (94)	194 (90)	177 (82)	0.001
Lifestyle					
Current smoker, *n* (%) ^k^	78 (12)	31 (14)	25 (12)	22 (10)	0.48
Alcohol intake >30 g/day, *n* (%) ^l^	29 (4)	9 (4)	11 (5)	9 (4)	0.60
SQUASH, intensity x hours	5050 (1950–8055)	5190 (1800–9105)	4750 (1700–7260)	5408 (2645–7301)	0.89
Diabetes and glucose homeostasis					
Diabetes mellitus, *n* (%)	160 (25)	50 (23)	58 (27)	52 (24)	0.53
Plasma glucose, mmol/L ^a^	5.2 (4.8–6.0)	5.3 (4.8–5.9)	5.2 (4.8–6.1)	5.3 (4.7–6.1)	0.09
HbA1c, % ^m^	5.8 (5.5–6.2)	5.7 (5.4–6.1)	5.8 (5.5–6.2)	5.9 (5.6–6.3)	0.004
Inflammation					
Leukocyte count, per 10^9^/L ^b^	8.2 ± 2.7	8.1 ± 2.8	8.2 ± 2.8	8.1 ± 2.4	0.97
hs-CRP, mg/L ^n^	1.6 (0.7–4.6)	1.7 (0.8–4.9)	1.4 (0.7–3.7)	1.6 (0.7–5.1)	0.71

Differences were tested by ANOVA for continuous variables with normal distribution, Kruskal–Wallis test for continuous variables with non-normal distribution and by χ^2^ test for categorical variables. Data available in ^a^ 647, ^b^ 648, ^c^ 637, ^d^ 649, ^e^ 633, ^f^ 636, ^g^ 623, ^h^ 638, ^i^ 608, ^j^ 567, ^k^ 607, ^l^ 581, ^m^ 627, ^n^ 613 patients. RTR, renal transplant recipients; uEGF, urinary epidermal growth factor; Cr, creatinine; eGFR, estimated glomerular filtration rate; SBP, systolic blood pressure; DBP, diastolic blood pressure; SQUASH, Short QUestionnaire to ASsess Health-enhancing physical activity; HDL, high–density lipoprotein cholesterol; LDL, low–density lipoprotein cholesterol; HbA1c, glycated hemoglobin; hs-CRP, high-sensitivity C-reactive protein.

**Table 2 jcm-08-01673-t002:** Multivariable-adjusted associations between uEGF/Cr and graft failure in 649 RTRs.

Models	uEGF/Cr, ng/mg		
	HR	95% CI	*P*
Crude	0.68	0.59–0.78	<0.001
Model 1	0.67	0.58–0.78	<0.001
Model 2	0.70	0.58–0.77	<0.001
Model 3	0.67	0.58–0.78	<0.001
Model 4	0.66	0.57–0.77	<0.001
Model 5	0.78	0.66–0.93	0.005
Model 6	0.79	0.67–0.94	0.007

In total 41 (6%) patients developed graft failure. Model 1: adjusted for age, sex, and transplant related data. Model 2: adjusted for age, sex, and renal transplant recipient characteristics. Model 3: Model 1 + Model 2. Model 4: adjusted for age, sex, and immunosuppressive therapy. Model 5: adjusted for age, sex, and eGFR and urinary protein excretion. Model 6: model 4 + model 5. RTRs, renal transplant recipients; uEGF, urinary epidermal growth factor.

**Table 3 jcm-08-01673-t003:** Predictive value (C-statistic) for uEGF/Cr on top of established risk factors for graft failure

	C-Statistic	*P* *
Urinary protein excretion, g/24 h	0.76	Ref.
+ uEGF/Cr, ng/mg	0.82	<0.001
Urinary albumin excretion, mg/24 h	0.78	Ref.
+ uEGF/Cr, ng/mg	0.82	<0.001

* *P*-value of F-test for difference between the reference model and the model plus uEGF/Cr. uEGF/Cr, urinary epidermal growth factor/creatinine ratio.

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
