# Peer review of "Urinary Epidermal Growth Factor/Creatinine Ratio and Graft Failure in Renal Transplant Recipients: A Prospective Cohort Study"

_jcm, 2019, doi:10.3390/jcm8101673_

Round 1

Reviewer 1 Report

Summary: The authors identify a well recognized and important question of long-term graft loss. Their research identified that EGF is associated with 32% less graft failure within a median of 5 years after transplantation. The authors' findings are interesting and have the potential to improve allograft function as they highlight a biomarker that may have a mechanistic role in GF. Below are some suggestions to the authors that will hopefully be helpful. 

Major comments

1) As the authors introduce their research question in the introduction, there is concern that there is an over simplistic view of long-term graft failure as it is described as mainly caused by chronic rejection, which leads to fibrosis. A focus on long-term graft failure being a multifaceted problem is more accurate. As the term "chronic allograft nephropathy" was specifically eliminated to emphasize that it is rather a poorly characterized histopathologic manifestation of IFTA rather than one diagnosis. It is a culmination of several factors with rejection being one of the factors, but also infection, immunosupression, hypertension, recipient characteristics, etc. The authors hypothesize that EGF may explain GF as it is part of the "unifying" pathway leading to fibrosis, which is a good hypothesis but it should be presented to the reader with clarity on the complexity of long-term GF. 

2) It would be helpful for the authors to show a study flow diagram. 

3) How many had missing GF data; were any imputations done? And how did authors deal with death prior to GF. Can results for death-censored GF also be provided. 

4) Authors use Jaffe to measure serum creatinine and limitations of using Jaffe should be mentioned in the limitations in the discussion. The presence of organic compounds such as ketones can give a false positive result. 

5) Can the authors mention the range of detection and inter and intra-assay CV for EGF. 

6) Under results, authors mention that higher EGF tertile received more acute rejection treatment, which conflicts with the data shown in Table 1. Please clarify. 

7) For figure 1, please show a table with number censored and number at risk at each interval follow up. 

8) For table 2, Model 1 should just be renamed "unadjusted" or crude. No need to state that subsequent models adjusted for model 1 as it is just the exposure of interest. it made it a bit confusing to follow. Can authors include a fully adjusted model with donor characteristics "transplant related data", and recipient characteristics.  There are some important donor characteristics that are missing from adjustment such as KDPI score or the variables making up KDPI including terminal serum creatinine, Hep C status, etc. Also information regarding cold ischemia time would be valuable to adjust for. Lastly, recipient  characteristics such as DGF, HLA  mismatch, PRA are also important to adjust for especially since authors describe their cohort as well phenotyped. Would be helpful to include these variables if available. If not, need to mention in limitations.

Minor comments

1) There are some spelling errors and grammatical errors throughout manuscript that will need to be addressed. examples: Unite States under affiliations should be United States. Also some sentences are incomplete such as the paragraph before the last in the methods section which has a sentence ending in 'were'. 

Author Response

Manuscript ID: jcm-583652

Detailed, itemized response to the comments of Reviewer #1.

Reviewer #1:

The authors identify a well-recognized and important question of long-term graft loss. Their research identified that EGF is associated with 32% less graft failure within a median of 5 years after transplantation. The authors' findings are interesting and have the potential to improve allograft function as they highlight a biomarker that may have a mechanistic role in GF. Below are some suggestions to the authors that will hopefully be helpful.

Response: We thank the Reviewer for the kind appraisal of our work. Please further see our responses to the comments listed below.

Specific comments:

Comment #1. As the authors introduce their research question in the introduction, there is concern that there is an over simplistic view of long-term graft failure as it is described as mainly caused by chronic rejection, which leads to fibrosis. A focus on long-term graft failure being a multifaceted problem is more accurate. As the term "chronic allograft nephropathy" was specifically eliminated to emphasize that it is rather a poorly characterized histopathologic manifestation of IFTA rather than one diagnosis. It is a culmination of several factors with rejection being one of the factors, but also infection, immunosuppression, hypertension, recipient characteristics, etc. The authors hypothesize that EGF may explain GF as it is part of the "unifying" pathway leading to fibrosis, which is a good hypothesis, but it should be presented to the reader with clarity on the complexity of long-term GF.

Response: We thank the Reviewer for the comment. We made several changes to the Introduction section of the revised version of the manuscript, as to provide a better appraisal of the multidirectional mechanisms involved. The introduction now reads: “Graft failure is a culmination of several factors, including chronic rejection, toxicity of calcineurin inhibitors, infection, hypertension, oxidative stress and proteinuria, together leading to progressive fibrosis and loss of renal function (Ponticelli C. Kidney Int 2000;75:S62–70, Schratzberger G et al. Nephrol Dial Transplant 2002;17(8):1384–1390, McLaren AJ et al. Ann Surg 2000;232(1):98–103) (Lines 35-37)”.  

Comment #2. It would be helpful for the authors to show a study flow diagram.

Response: In agreement with the Reviewer, in the revised version of the manuscript we now include the following flow diagram (Figure 1 in the revised version of the manuscript, see also below), which is referenced in subheading 2.1 Study design and population of the Materials and Methods section.

Comment #3. How many had missing GF data; were any imputations done? And how did authors deal with death prior to GF. Can results for death-censored GF also be provided.

Response: We thank the Reviewer for this comment. There were no missing GF data. No imputations were performed. Patients that died with a functioning graft were censored at time of death. We show the results of Cox regression analyses with the endpoint death-censored graft-failure. To accommodate this comment of the Reviewer, in the Materials and Methods section of the revised version of the manuscript, we now explain the aforementioned as following: "The primary end point of the current study was death-censored graft failure (Line 71)" and “No participants were lost to follow-up (Line 74)”.

Comment #4. Authors use Jaffe to measure serum creatinine and limitations of using Jaffe should be mentioned in the limitations in the discussion. The presence of organic compounds such as ketones can give a false positive result.

Response: We agree with the Reviewer. Following the recommendation by the Reviewer we now recognize this limitation in the Discussion section of the revised version of the manuscript, which now reads as follows: “Moreover, we used the Jaffé method to measure serum creatinine, which can generate false positive results in the presence of pesudochromogens such as ketones (Lee E et al. Clin J Am Soc Nephrol 2017;12(1):29–37) (Lines 281-283)”.

Comment #5. Can the authors mention the range of detection and inter and intra-assay CV for EGF.

Response: We thank the Reviewer for this comment. In subheading 2.3 Laboratory measurements and calculations of the Materials and Methods section of the revised version of the manuscript we now mention the range of detection and inter and intra-assay CV for EGF, which reads as following: “The test has a range of detection of 3.9-250 pg/mL; the intra- and inter-plate coefficients of variation were less than 10% and 15%, respectively (Ju W et al. Sci Transl Med 2016;7(316): 316ra193) (Lines 97-98)".

Comment #6. Under results, authors mention that higher EGF tertile received more acute rejection treatment, which conflicts with the data shown in Table 1. Please clarify.

Response: We thank the Reviewer for this comment. In subheading 3.1 Characteristics at enrollment of the Results section of the revised version of the manuscript we now provide the corrected sentence which reads as following: “a smaller percentage of patients required acute rejection treatment (P<0.001) (Lines 158-159)".

Comment #7. For figure 1, please show a table with number censored and number at risk at each interval follow up.

Response: We thank the Reviewer for this comment. In the revised version of the manuscript, we now provide a Kaplan Mayer curve (now presented as Figure 2, due to incorporation of the Flow Diagram as explained in comment #2) with addition of a table with the information regarding number censored and number at risk at each interval follow-up. Figure 2 is now depicted in the manuscript as follows:

Comment #8. For table 2, Model 1 should just be renamed "unadjusted" or crude. No need to state that subsequent models adjusted for model 1 as it is just the exposure of interest. it made it a bit confusing to follow. Can authors include a fully adjusted model with donor characteristics "transplant related data", and recipient characteristics.  There are some important donor characteristics that are missing from adjustment such as KDPI score or the variables making up KDPI including terminal serum creatinine, Hep C status, etc. Also information regarding cold ischemia time would be valuable to adjust for. Lastly, recipient  characteristics such as DGF, HLA  mismatch, PRA are also important to adjust for especially since authors describe their cohort as well phenotyped. Would be helpful to include these variables if available. If not, need to mention in limitations.

Response: In agreement with the Reviewer, in the revised version of the manuscript we have renamed "model 1" to "unadjusted", and we have incorporated several additional variables to the multivariable-adjusted Cox regression models, as shown in Table 2. In detail, we added adjustment for cold ischemia time in model 1 (transplant characteristics-related model); we added a new model that includes recipient characteristics, namely: delayed graft function and number of HLA mismatches (Model 2); and we merged Model 1 and Model 2 (Model 3). Of note, the prospective association between uEGF/Cr and graft failure remained independent of these adjustments. Because we did not have available the requested information regarding donor characteristics, we acknowledge this limitation in the Discussion section of the revised version of the manuscript, which reads as following: “Next, only limited data were available regarding donors characteristics and therefore we could not adjust for donor variables such as donor serum creatinine or donor Hepatitis C status (Lines 283-285)”.

Comment #9. There are some spelling errors and grammatical errors throughout manuscript that will need to be addressed. examples: Unite States under affiliations should be United States. Also some sentences are incomplete such as the paragraph before the last in the methods section which has a sentence ending in 'were'.

Response: We thank the Reviewer for noting this. In the revised version of the manuscript, we have made several corresponding spelling and grammar amendments in Lines 10, 46, 53, 97, 190, 243.

References

Ponticelli, C. Progression of renal damage in chronic rejection. Kidney Int 2000;75:S62–70.

Schratzberger G.; Gert, M. Chronic allograft failure: a disease we don't understand and can't cure? Nephrol Dial Transplant 2002;17(8):1384–1390.

McLaren, A.J.; Fuggle, S.V.; Welsh, K.I.; Gray, D.W.R.; Phil, D.; Morris, P.J. Chronic Allograft Failure in Human Renal Transplantation. A Multivariate Risk Factor Analysis. Ann Surg 2000, 232(1), 98–103.

Lee, E.; Collier, C.P.; White, C.A. Interlaboratory Variability in Plasma Creatinine Measurement and the Relation with Estimated Glomerular Filtration Rate and Chronic Kidney Disease Diagnosis. Clin J Am Soc Nephrol 2017, 12(1), 29–37.

Ju, W.; Nair, V.; Smith, S.; Zhu, L.; Shedden, K.; Song, P.X.K.; Mariani, L.H.; Eichinger, F.H.; Berthier, C.C.; Randolph, A.; et al. Tissue transcriptome-driven identification of epidermal growth factor as a chronic kidney disease biomarker. Sci Transl Med 2015, 7(316), 316ra193.

Reviewer 2 Report

Yepes-Calderon et al. investigated the potential of urinary EGF/creatinine as a biomarker of graft failure after kidney transplantation. It is a well-written paper, which shows robust and interesting results. An important limitation of the study is – as mentioned by the authors in the discussion section – that the prediction models are based on a single EGF measurement. Since this was a prospective study, it would have been of tremendous interest to include multiple sampling times. However, the results are interesting. There remain, however, a few remarks as described below that need to be addressed.

The authors mention “baseline data” and “baseline characteristics”. Why is it called “baseline”. There is no accordance between patients since the time after transplantation differs between 1.74 and 12 years after transplantation and there is no event defining the decision of sampling. I suggest to adapt the phrase ‘baseline” to a more suitable term. Are EGF values in your population of kidney transplant recipients comparable to values described in literature? How do they relate to healthy subjects? As the authors probably know, EGF is also involved in the renal magnesium reabsorption in the distal convoluted tube (PMID:17671655). It has been shown that urinary EGF decreases with the use of calcineurin inhibitors after kidney transplantation in adults and in children (PMID: 24353324 and PMID: 29861470). In the present study, the authors indeed found that the use of calcineurin inhibitors is the highest in the low EGF group, which is in accordance with the existing evidence. The authors should add a paragraph on this topic in the discussion section and explain why they think their findings are independent from the use of CNI. How do the authors explain that the time after transplantation is similar in the 3 groups? The risk on graft failure increases with time after transplantation (PMID: 31530561). One could speculate that EGF decreases with time after transplantation and decreasing kidney function. How was body surface area calculated? The authors did not find a difference in the prevalence of diabetes mellitus in the 3 groups. It has been published, however, that EGF is lower in patients with diabetes mellitus (PMID: 2286016). Can the authors comment on that in the discussion? It would be interesting to repeat the analysis to compare the level of HbA1c in the 3 groups when only including patients with diabetes mellitus. The authors should add the positive predictive value and the negative predictive value of EGF as a biomarker to predict graft failure. Figure 2: please change the colours of the 3 ROC lines so that the difference between the 3 is clear. Figure 2: did the authors check if EGF might be of additional value to urinary albumin and/or protein excretion? Combining 2 or 3 of them could results in an increased predictive model.

Author Response

Manuscript ID: jcm-583652

Detailed, itemized response to the comments of Reviewer #2.

Reviewer #2:

Yepes-Calderon et al. investigated the potential of urinary EGF/creatinine as a biomarker of graft failure after kidney transplantation. It is a well-written paper, which shows robust and interesting results. An important limitation of the study is – as mentioned by the authors in the discussion section – that the prediction models are based on a single EGF measurement. Since this was a prospective study, it would have been of tremendous interest to include multiple sampling times. However, the results are interesting. There remain, however, a few remarks as described below that need to be addressed.

Response: We thank the Reviewer for the kind appraisal of our work. Please further see our responses to the comments listed below.

Specific comments:

Comment #1. The authors mention “baseline data” and “baseline characteristics”. Why is it called “baseline”. There is no accordance between patients since the time after transplantation differs between 1.74 and 12 years after transplantation and there is no event defining the decision of sampling. I suggest to adapt the phrase ‘baseline” to a more suitable term.

Response: In agreement with the Reviewer, in the revised version of the manuscript we have replaced the term “Baseline” by “Enrollment” throughout the manuscript.

Comment #2. Are EGF values in your population of kidney transplant recipients comparable to values described in literature? How do they relate to healthy subjects?

Response: We thank the Reviewer for the comment. Reported uEGF/Cr ratios in healthy subjects range from 18 to 25 ng/mg (Meybosch S et al. PLoS One 2019;14:e0211212, Callegari C et al. Eur J Appl Physiol Occup Physiol 1988;58(1-2):26–31). It has consistently been found that patients with chronic kidney disease (CKD) have lower uEGF/Cr ratios than healthy subjects, with values ranging from a mean of 5.65 ng/mg in the C-PROBE cohort of patients with CKD, to a mean of 11.37 ng/mg in patients with IgA nephropathy and mild renal function impairment (Ju W et al. Sci Transl Med 2016;7(316):316ra193). In agreement with those reports, the median value of uEGF/Cr in our population (6.43 ng/mg) shows to be lower than the healthy population and in the range of values previously found in patients with CKD. We highlight this point in the Discussion section of the revised version of the manuscript, which reads as following: “EGF has gained interest as a biomarker of renal disease because its decreased urinary excretion has been observed in nearly all rodent kidney injury models and in various human kidney diseases, including diabetic nephropathy, IgA nephropathy and lupus nephritis (Isaka Y. Ann Transl Med 2016;4(Suppl 1):S62). Consistently, we found that our study population of RTR had a decreased uEGF/Cr ratio when compared to healthy subjects, and comparable ratios to those of patients with CKD (Ju W et al. Sci Transl Med 2016;7(316):316ra193, Meybosch S et al. PLoS One 2019;14:e0211212, Callegari C et al. Eur J Appl Physiol Occup Physiol 1988;58(1-2):26–31) (Lines 239-244)”.

Comment #3. As the authors probably know, EGF is also involved in the renal magnesium reabsorption in the distal convoluted tube (PMID:17671655). It has been shown that urinary EGF decreases with the use of calcineurin inhibitors after kidney transplantation in adults and in children (PMID: 24353324 and PMID: 29861470). In the present study, the authors indeed found that the use of calcineurin inhibitors is the highest in the low EGF group, which is in accordance with the existing evidence. The authors should add a paragraph on this topic in the discussion section and explain why they think their findings are independent from the use of CNI.

Response: We thank the Reviewer for the comment. In agreement, we added the following paragraph to the discussion section of the revised version of the manuscript: “We also found in our population that the use of calcineurin inhibitors was higher among patients with lower uEGF/Cr. This is in agreement with previous studies showing an inverse association between uEGF and the use of calcineurin inhibitors (Ledeganck KJ et al. Nutrients 2018;10(6):E677, Ledeganck KJ et al. Nephrol Dial Transplant 2014;29(5):1097–1102) and a potential involvement of the EGF receptor in the alterations that lead to magnesium loss in renal transplant recipients receiving calcineurin inhibitors (Ledeganck KJ et al. Nephrol Dial Transplant 2014; 29(5):1097–1102, Groenestege WM et al. J Clin Invest 2007;117(8):2260–2267). Nevertheless, the association between uEGF/Cr and graft failure was independent of the adjustment for the use of calcineurin inhibitors. This suggests that the association of uEGF/Cr is not mediated by a nephrotoxic effect of calcineurin inhibitors, but is mediated by other mechanisms, which may involve renal fibrosis. (Lines 263-270)”.

Comment #4.  How do the authors explain that the time after transplantation is similar in the 3 groups? The risk on graft failure increases with time after transplantation (PMID: 31530561). One could speculate that EGF decreases with time after transplantation and decreasing kidney function.

Response: We thank the Reviewer for the comment. To accommodate the comment of the Reviewer, we performed linear regression analyses to test the association between time after transplantation and uEGF/Cr in crude and multivariable models with adjustment for use of cyclosporin inhibitors. In crude analyses we did not find signs of a significant association (Std. B=-0.015; P=0.71), which is in line with the results of analysis of variance across tertiles of uEGF shown in Table 1. Whilst, the inverse association between uEGF and time after transplantation, as suggested by the Reviewer, was detected after adjustment for cyclosporine use (Std. B=0.81; P=0.046). In the Methods and Results sections of the revised version of the manuscript, we now mention performance of these analyses (Lines 115-117) and report the results (Lines 149-152). Finally, in the Discussion section of the revised version of the manuscript, we provide interpretation of these results as following: "Because risk of graft failure increases with time, one could speculate that uEGF decreases with time after transplantation. However, we did not observe such a relationship over increasing tertiles of uEGF. This finding may be explained by a confounding effect of use of cyclosporine  resulting in lowering of uEGF, which is supported by the observation that an association between uEGF and time after transplantation became apparent after adjustment for use of cyclosporine in linear regression analyses (Lines 258-263)”.

Comment #5. How was body surface area calculated?

Response: We thank the Reviewer for this comment. Accordingly, in subheading 2.3. Laboratory measurements and calculations of the Materials and Methods section of the revised version of the manuscript we now detailed: “Body surface area was calculated according to the Du Bois formula (Du Bois D et al. Nutrition 1989;5(5):303–311) (Line 102)”.

Comment #6. The authors did not find a difference in the prevalence of diabetes mellitus in the 3 groups. It has been published, however, that EGF is lower in patients with diabetes mellitus (PMID: 2286016). Can the authors comment on that in the discussion?

Response: We thank the Reviewer for the comment. To accommodate this comment, in the Discussion section of the revised version of the manuscript, we have added the following paragraph commenting on this: “Other than in a previous study (Lev-Ran A et al. Clin Chim Acta 1990;192(3):201–216), in our population no difference in prevalence of diabetes was found over tertiles of uEGF/Cr ratio (Lines 271-272)”.

Comment #7. It would be interesting to repeat the analysis to compare the level of HbA1c in the 3 groups when only including patients with diabetes mellitus.

Response: In agreement with the Reviewer, in the revised version of the manuscript we have repeated the analysis comparing HbA1c percentage in both diabetic and non-diabetic subjects. To accommodate this comment, the Results section of the revised version of the manuscript now reads as following: “Patients in the higher uEGF/Cr tertile also had higher glycosylated hemoglobin percentage, (Table 1) independent of whether they were diabetic or non-diabetic subjects (Table S2) (Lines 159-160)”.

Comment #8. The authors should add the positive predictive value and the negative predictive value of EGF as a biomarker to predict graft failure.

Response: We thank the Reviewer this comment. We calculated the positive predictive value and negative predictive value based on tertiles of uEGF/Cr distribution. We have added results to the revised version of the manuscript, which now reads: “Being in the first tertile of uEGF/Cr had a positive predictive value of 75% for the development of graft failure, whereas being in the third tertile had a negative predictive value of 81% (Lines 187-189)”. The complete report of these analyses is now shown in the supplemental material (Table S3 of the revised version of the Supplemental Material, see also below).

Table S3. Positive and negative predictive value by tertiles of uEGF/Cr.

Tertiles of uEGF/cr

Tertile 1

Tertile 3

Cut-off value

< 4.47 ng/mg

> 8.81 ng/mg

Positive predictive value, (%)

75

58

Negative predictive value, (%)

78

81

uEGF/Cr, urinary epidermal growth factor/creatinine ratio.

Comment #9. Figure 2: please change the colours of the 3 ROC lines so that the difference between the 3 is clear.

Response: We thank the Reviewer for this comment. In agreement, in the revised version of the manuscript, we have changed the color and pattern settings of the ROC lines, and we increased the size of Figure 2 to allow unequivocal visualization.

Comment #10. did the authors check if EGF might be of additional value to urinary albumin and/or protein excretion? Combining 2 or 3 of them could results in an increased predictive model.

Response: We thank the Reviewer for the comment. In the Methods section of the revised version of the manuscript we mention the performance of this analyses, which reads as follows: "To investigate whether uEGF might be of additional value to urinary albumin excretion and protein excretion, we calculated the individual C-statistics of these variables, and then the C-statistics of them combined with uEGF/Cr. Moreover, we performed an F-test to check whether the difference between predictive models was significant (Lines 134-137)”. In the Results section of the revised version of the manuscript, we show that adding uEGF/Cr to either urinary albumin excretion or urinary protein excretion significantly improves prediction of graft failure (see new Table 3, also below)

Table 3. Predictive value (C-statistic) for uEGF/Cr on top of established risk factors for graft failure

C-statistic

P*

Urinary protein excretion, g/24h

0.76

Ref.

+uEGF/Cr, ng/mg

0.82

<0.001

Urinary albumin excretion, mg/24h

0.78

Ref.

+uEGF/Cr

0.82

<0.001

*P-value of F-test for difference between the reference model and the model plus uEGF/Cr. uEGF/Cr, urinary epidermal growth factor/creatinine ratio.

References

Meybosch, S.; De Monie, A.; Anné, C.; Bruyndonckx, L.; Jürgens, A.; De Winter, B.Y.; Trouet, D.; Ledeganck, K.J. Epidermal growth factor and its influencing variables in healthy children and adults. PLoS One 2019, 14, e0211212.

Callegari, C.; Laborde, N.P.; Buenaflor, G.; Nascimento, C.G.; Brasel, J.A.; Fisher, D.A. The source of urinary epidermal growth factor in humans. Eur J Appl Physiol Occup Physiol 1988, 58(1-2), 26–31.

Ju, W.; Nair, V.; Smith, S.; Zhu, L.; Shedden, K.; Song, P.X.K.; Mariani, L.H.; Eichinger, F.H.; Berthier, C.C.; Randolph, A.; et al. Tissue transcriptome-driven identification of epidermal growth factor as a chronic kidney disease biomarker. Sci Transl Med 2016, 7(316), 316ra193.

Isaka, Y. Epidermal growth factor as a prognostic biomarker in chronic kidney diseases. Ann Transl Med 2016, 4(suppl 1), S62.

Ledeganck, K.J.; Anné, C.; De Monie, A.; Meybosch, S.; Verpooten, G.; Vinckx, M.; Van Hoeck, K.; Van Eyck, A.; De Winter, B.; Trouet, D. Longitudinal Study of the Role of Epidermal Growth Factor on the Fractional Excretion of Magnesium in Children: Effect of Calcineurin Inhibitors. Nutrients 2018, 10(6), E677.

Ledeganck, K.J.; De Winter, B.Y.; Van den Driessche, A.; Jurgens, A.; Bosmans, J.-L.; Couttenye, M.M.; Verpooten, G.A. Magnesium loss in cyclosporine-treated patients is related to renal epidermal growth factor downregulation. Nephrol Dial Transplant 2014, 29(5), 1097–1102.

Groenestege, W.M.; Thébault, S.; van der Wijst, J.; van den Berg, D.; Janssen, R.; Tejpar, S.; van den Heuvel, L.P.; van Cutsem, E.; Hoenderop, J.G.; Knoers, N. V.; et al. Impaired basolateral sorting of pro-EGF causes isolated recessive renal hypomagnesemia. J Clin Invest 2007, 117(8), 2260–2267.

Du Bois, D.; Du Bois, E.F. A formula to estimate the approximate surface area if height and weight be known. Nutrition 1989, 5(5), 303–311.

Lev-Ran, A.; Hwang, D.L.; Miller, J.D.; Josefsberg, Z. Excretion of epidermal growth factor (EGF) in diabetes. Clin Chim Acta 1990, 192(3), 201–206.

Round 2

Reviewer 2 Report

The authors satisfactorily responded to the comments. A few spell errors should be corrected:

r59: phenomenon

r85: data were collected

r155: transplantation and uEGF/Cr

r156: adjustment for calcineurin inhibitor

r285: a reformulation of this sentence would make it more readable.  For example: In contrast to previous results [31], no difference in the prevalence of diabetes was observed between the uEGF/c tertiles in our study population.

Author Response

Manuscript ID: jcm-583652

Detailed, itemized response to the comments of Reviewer #2.

Reviewer #2:

The authors satisfactorily responded to the comments. A few spell errors should be corrected:

r59: phenomenon

r85: data were collected

r155: transplantation and uEGF/Cr

r156: adjustment for calcineurin inhibitor

r285: a reformulation of this sentence would make it more readable.  For example: In contrast to previous results [31], no difference in the prevalence of diabetes was observed between the uEGF/c tertiles in our study population.

Response: We thank the Reviewer for the kind appraisal and thoughtful revision of our work. We have now corrected the spelling errors in the revised version of the manuscript (Rows 59, 85, 154, 156, and 157). Also, in agreement with the Reviewer, we have adapted the sentence in the Discussion section of the revised manuscript, which now reads: “in contrast to previous results,[31] no difference was observed in the prevalence of diabetes between the uEGF/Cr tertiles in our study population (Rows 286-288)”.
